# Effect of COVID-19 on Thoracic Oncology Surgery in Spain: A Spanish Thoracic Surgery Society (SECT) Survey

**DOI:** 10.3390/cancers13122897

**Published:** 2021-06-09

**Authors:** Néstor J. Martínez-Hernández, Usue Caballero Silva, Alberto Cabañero Sánchez, José Luis Campo-Cañaveral de la Cruz, Andrés Obeso Carillo, José Ramón Jarabo Sarceda, Sebastián Sevilla López, Ángel Cilleruelo Ramos, José Luis Recuero Díaz, Sergi Call, Felipe Couñago, Florentino Hernando Trancho

**Affiliations:** 1Thoracic Surgery, Hospital Universitari de la Ribera, 46600 Alzira, Spain; 2Thoracic Surgery, Hospital Universitario Ramón y Cajal, 28034 Madrid, Spain; usue.caballero@salud.madrid.org (U.C.S.); Alberto.cabanero@salud.madrid.org (A.C.S.); 3Thoracic Surgery, Hospital Universitario Puerta de Hierro-Majadahonda, 28222 Majadahonda, Spain; campo-canaveral.delacruz@salud.madrid.org; 4Thoracic Surgery, Hospital Clínico Universitario de Santiago, 15706 Santiago de Compostela, Spain; gerardo.andres.obeso.carillo@sergas.es; 5Thoracic Surgery, Hospital Clínico San Carlos, 28040 Madrid, Spain; joseramon.jarabo@salud.madrid.org (J.R.J.S.); florentino.hernando@salud.madrid.org (F.H.T.); 6Thoracic Surgery, Hospitalario Universitario Ciudad de Jaén, 23007 Jaén, Spain; sebastian.sevilla.sspa@juntadeandalucia.es; 7Thoracic Surgery, Hospital Clínico de Valladolid, 47003 Valladolid, Spain; acilleruelo@saludcastillayleon.es; 8Thoracic Surgery, Instituto de Investigación Sanitaria de Aragón, Hospital Universitario Miguel Servet, 50009 Zaragoza, Spain; jlrecuero@salud.aragon.es; 9Thoracic Surgery, Hospital Universitari Mútua Terrassa, 08221 Terrassa, Spain; scall@mutuaterrassa.cat; 10Radiation Oncology, Hospital Universitario Quirónsalud, 28223 Pozuelo de Alarcón, Spain; felipe.counago@quironsalud.es; 11Radiation Oncology, Hospital La Luz, 28003 Madrid, Spain; 12Radiation Oncology, Universidad Europea de Madrid, 28670 Villaviciosa de Odón, Spain

**Keywords:** lung cancer, surgical treatment, COVID-19

## Abstract

**Simple Summary:**

After the first wave of COVID-19, the Spanish Society of Thoracic Surgeons (SECT) surveyed its members to assess the impact of the pandemic on thoracic oncology surgery in one of the counties most affected by the virus. In May 2020, all SECT members were invited to complete a 40-item, multiple choice questionnaire by e-mail. The response rate was 19.0%. Surgical activity decreased by 95.7%, with 41.5% of centers performing surgery only in oncologic cases and 11.7% only in emergencies. More than half (56%) of multidisciplinary tumour board meetings (56%) were cancelled or conducted online. Standard protocols for early-stage disease were modified in 62.9% of centers. The results of this survey show that the COVID-19 pandemic severely limited thoracic oncology surgery activity. Here we describe and discuss the impact of the pandemic on thoracic surgery in Spain.

**Abstract:**

After the first wave of COVID-19, the Spanish Society of Thoracic Surgeons (SECT) surveyed its members to assess the impact of the pandemic on thoracic oncology surgery in Spain. In May 2020, all SECT members were invited to complete an online, 40-item, multiple choice questionnaire. The questionnaire was developed by the SECT Scientific Committee and sent via email. The overall response rate was 19.2%. The respondents answered at least 91.5% of the items, with only one exception (a question about residents). Most respondents (89.3%) worked in public hospitals. The reported impact of the pandemic on routine clinical activity was considered extreme or severe by 75.5% of respondents (25.5% and 50%, respectively). Multidisciplinary tumour boards were held either with fewer members attending or through electronic platforms (44.6% and 35.9%, respectively). Surgical activity decreased by 95.7%, with 41.5% of centers performing surgery only on oncological patients and 11.7% only in emergencies. Nearly 60% of respondents reported modifying standard protocols for early-stage cancer and in the preoperative workup. Most centers (≈80%) reported using full personal protective equipment when operating on COVID-19 positive patients. The COVID-19 pandemic severely affected thoracic oncology surgery in Spain. The lack of common protocols led to a variable care delivery to lung cancer patients.

## 1. Introduction

Lung cancer is the most lethal type of cancer, accounting for 1.76 million deaths annually [1]. The treatment of lung cancer is multimodal and multiple strategies are available depending on the stage at diagnosis. Approximately 25–30% of patients are diagnosed with early-stage disease [2]. In many of these patients, lung resection is the treatment of choice [3]. Although lung cancer is often considered an epidemic, the emergence of a new coronavirus variant in December 2019—SARS-CoV-2, which causes the disease known as COVID-19—quickly eclipsed lung cancer and most other health conditions. A few months later, on 11 March 2020, the World Health Organization officially declared a pandemic [4,5]. In only one year, COVID-19 has directly or indirectly caused over 3 million deaths worldwide [6], forcing governments around the world to implement strict measures restricting the free movement of citizens and bringing the economy to a halt [7,8,9].

The last major pandemic was the flu of 1918 [10] when the field of medicine and hospital organisation were very different from the present. Moreover, at that time, the limited therapeutic arsenal for lung cancer did not yet include surgery [11], which explains the unprecedented impact of the current pandemic on the diagnosis and treatment of lung cancer.

In Spain, the epidemiological situation worsened quickly after the initial outbreak of the virus, reaching nearly catastrophic proportions. In the first wave, Spain had the highest excess mortality rate per 100,000 inhabitants in Europe and also the highest relative excess number of deaths, which was only surpassed among males in certain regions of the United Kingdom [12]. Strict measures were imposed to restrict the movement of the population during the early months of the pandemic from March to May, 2020. Nevertheless, COVID-19 patients accounted for more than 100% of hospital occupancy (105%) in some regions, which required the addition of more beds in cafeterias, libraries, gyms, etc. In some cases, intensive care unit (ICU) occupancy rates were as high as 300% [13], and field hospitals were created in several cities to accommodate patient overflow [14]. Hospitals were overwhelmed during this first wave, which had a major impact on the treatment of all non-COVID-19 conditions. In many cases—and lung cancer was no exception—surgery had to be cancelled or postponed indefinitely [15].

In this context, in which operating room availability was greatly reduced or even completely unavailable due to closures, the main national and international scientific societies issued recommendations for the management of patients with thoracic cancer. Those recommendations called for triaging patients for surgery based on the theoretical deferability of the operation, taking into account the safety of patients and surgical teams alike, and adapting the recommendations to the specific conditions in each region or country [16,17]. These guidelines, together with the use of COVID-19-free areas of the hospital, were particularly important in Spain, in which the high incidence of SARS-CoV-2 and consequent hospital overload had a major impact on surgical procedures. Importantly, this strategy allowed for the surgical treatment of selected patients with little to no excess morbidity and mortality [18,19].

To evaluate the true impact of the pandemic on thoracic surgery departments in a country powerfully affected by the first wave of the COVID-19 pandemic, the Spanish Society of Thoracic Surgery (SECT) carried out an anonymous survey to obtain first-hand information from its members. The main aim of the present study is to report and discuss the results of that survey to provide insight into the treatment of lung cancer during one of the most adverse scenarios imaginable. The survey had three main objectives regarding the surgical treatment of lung cancer in Spain during the first wave of COVID-19: (1) to audit the activity of the multidisciplinary tumor board (MTB) and decision-making during this challenging period; (2) to determine the extent to which the pandemic affected the preoperative diagnostic pathway; and (3) to determine the impact on the surgical procedures and protocols.

## 2. Materials and Methods

Members of the SECT, the association that represents all thoracic surgeons in Spain, were invited by email to participate in this survey. The first invitation was sent on 7 May 2020 to the 471 members of the SECT. The survey was created on the SurveyMonkey platform (www.surveymonkey.co.uk, accessed on 5 May 2020). Reminders were subsequently sent by email on 12 May and 20 May. The survey remained open until 29 May 2020.

The survey was designed by the SECT Scientific Committee and consisted of 40 multiple choice questions. Of these, nine questions were designed to assess the members’ sociodemographic background and the general operation of their hospital. There were 28 questions on the impact of the pandemic on the surgical treatment of thoracic cancer, two on the quality of the scientific studies on COVID-19 published to date, and one on the impact of the pandemic on training of thoracic surgeons. The 28 questions about the impact on surgery focused on the following areas: functioning of the MTB and waiting lists (five questions); preoperative studies (four questions); aspects related to the surgical procedure, postoperative care, and the management of pleural fluids (10 questions); SARS-CoV-2 diagnosis and testing in patients scheduled for surgery (five questions); and protective measures for physicians (four questions). The estimated time to complete the survey was nine minutes. The complete survey is shown in Table 1.

### Statistical Analysis

A descriptive analysis of the data obtained from the survey was performed. All results are given as absolute numbers and percentages.

## 3. Results

A total of 471 SECT members were surveyed and 94 completed the survey, for an overall response rate (RR) of 19.2%.

### 3.1. Sociodemographic and Hospital-Related Data

Sociodemographic and hospital-related data are shown in Table 2 and Figure 1.

### 3.2. Multidisciplinary Teams and Cancer

**Question** **10.**
*Was it possible to maintain the routine work of the multidisciplinary tumour board at your hospital? (RR: 92/94; 97.8%).*


In most of the participating centers, MTB meetings continued to be held during the pandemic. In 44.6% of centers (*n* = 41), these meetings were held in person but with a reduced number of attendees with distancing to avoid close contact. In 35.9% of centers (*n* = 33), the meetings were held through electronic platforms. By contrast, 18.5% of centers (*n* = 17) completely cancelled all MTB meetings (Figure 2A).

**Question** **11.**
*In patients with early-stage lung cancer, did the pandemic alter the treatment decisions made by the MTB? (RR: 91/94; 96.8%).*


Nearly 40% of respondents (*n* = 36, 39.6%) reported that the pandemic did not influence the management of patients with early-stage lung cancer. However, 47.3% (*n* = 43) referred more patients to surgery while 9.9% (*n* = 9) referred more patients to radiation therapy (Figure 2B).

**Question** **12.**
*In patients with locally-advanced lung cancer, did the pandemic alter the treatment decisions made by the MTB? (RR: 90/94; 95.7%).*


Most centers did not modify the management of patients with locally-advanced disease (*n* = 60, 66.7%). However, in 28.9% of centers (*n* = 26), more patients were prescribed chemotherapy. One center (1.1%) reported that surgery was indicated in more patients (Figure 2C).

**Question** **13.**
*What is your opinion regarding the changes in the management of patients with lung cancer? (RR: 92/94; 97.8%).*


Slightly more than half of respondents (53.3%; *n* = 49) agreed with these modifications. By contrast, 43.5% (*n* = 40) believed that patients should have been referred to COVID-19-free centers for treatment.

**Question** **14.**
*Since the start of the pandemic, what is the average waiting time for surgery in your cancer patients? (RR: 93/94; 98.9%).*


Twenty-nine centers (31.2%) reported that waiting time for surgery was less than one month. However, in most centers (*n* = 47, 50.5%) waiting times ranged from 1 to 2 months. In the remaining centers (*n* = 16, 172%), the waiting time was 2–3 months.

### 3.3. Patient Screening

Table 3 summarises the findings regarding patient screening.

### 3.4. Preoperative Workup

**Question** **20.**
*How has the COVID-19 pandemic influenced the preoperative workup? (RR: 92/94; 97.8%)*


A total of 37 respondents (40.2%) responded that preoperative workup was unchanged from the pre-pandemic period. Among the centers that reported changes, the preoperative tests that were most affected (i.e., delays and/or omissions) were: bronchoscopy (56.5% of centers), CT-guided biopsy (42.4%), and referral to the pneumology department for assessment of lung nodules (29.4%).

**Question** **21.**
*Which of the following preoperative lung function tests (pulmonary function test, pulmonary diffusion test …) are not available due to the COVID-19 pandemic? (RR: 94/94; 100%)*


Forty participants (42.5%) responded that all tests remained the same. Fifty respondents (53.2%) indicated that spirometry and pulmonary diffusion tests were unavailable or with reduced availability, while 36 (38.3%) reported a delay or unavailability of positron-emission tomography (PET) imaging. Sixteen (17%) respondents indicated that the pandemic negatively impacted the availability of ventilation-perfusion lung scintigraphy.

**Question** **22.**
*What is your opinion with regard to changes in the preoperative workup? (RR: 94/94; 100%)*


Most participants (*n* = 81, 86.2%) agreed with the preoperative tests that were performed. Only one (1.1%) thought that fewer tests should be carried out while 12 (12.8%) believed that more tests should be performed.

**Question** **23.**
*How has the pandemic affected consultations in thoracic surgery? (RR: 94/94; 100%)*


Only one respondent (1.1%) indicated that the center maintained the same activity level. Three centers (3.2%) cancelled all consultations. In most centers (*n* = 76, 80.8%) the initial consultation and the first postoperative consultation were maintained, while the remaining consultations were performed online. Finally, 14 centers (14.9%) reported that all consultations were performed through electronic platforms.

### 3.5. Surgery

**Question** **24.**
*How has the pandemic affected surgical activity in your department? (RR: 94/94; 100%)*


Four respondents (4.3%) reported that the activity level was unchanged while 3 (3.2%) cancelled all surgical interventions. A total of 37 respondents (39.4%) suspended surgical treatment for benign tumours, and 39 (41.5%) only performed surgery in cancer patients. Eleven centers (11.7%) performed only emergency surgeries.

**Question** **25.**
*What is your opinion with regard to these changes in surgical interventions? (RR: 94/94; 100%)*


Most participants (*n* = 59, 62.8%) agreed with the changes made and two respondents (2.1%) believed that this activity should be further reduced. By contrast, 17 respondents (18.1%) disagreed with the changes because this implied suboptimal patient care. Sixteen respondents (17%) believed that more surgical interventions should be performed.

**Question** **26.**
*Has the postoperative length of stay in the ICU/recovery unit been affected by the pandemic? (RR: 93/94; 98.9%)*


Most respondents (*n* = 64, 68.8%) reported that the length of stay remained unchanged. However, 19 (20.4%) indicated that the stays were shorter while 10 (10.8%) cancelled postoperative stays.

**Question** **27.**
*Were any of the patients admitted to your department (regardless of surgical status) diagnosed with SARS-CoV-2? (RR: 93/94; 98.9%)*


Most of the participants, 52 (55.9%) reported having no infected patients in their department. Ten centers (10.7%) reported one positive case, 27 centers (29%) reported <5, and four centers (4.3%) >5 cases.

**Question** **28.**
*What recommendations did you use for surgical planning in your department? (RR: 93/94; 98.9%)*


Thirty-three respondents (35.5%) based surgical planning on the recommendations of the American College of Surgeons (ACS). More than half of respondents (*n* = 47, 50.5%) based surgical planning on common sense. Eight centers halted all activity and thus patient prioritisation was not necessary. Five centers (5.4%) based surgical planning on links published on the SECT website (links to studies and international guidelines).

**Question** **29.**
*If surgical activity has continued at your hospital, have you observed any increase in morbidity and/or mortality? (RR: 91/94 96.8%)*


Most respondents (*n* = 65, 75.6%) reported that morbidity and mortality rates were unchanged. However, five respondents (5.8%) reported a higher morbidity and mortality rates due to the reduced availability of both material and human resources. Five respondents (5.8%) reported an increase in morbidity unrelated to the pandemic while five other centres (5.8%) reported an increase in morbidity associated with COVID-19.

**Question** **30.**
*What types of surgical interventions (if any) have been performed in your department on COVID-19 patients? (multiple choice response) (RR: 90/94; 95.7%)*


Seven respondents (7.8%) indicated that no surgical procedures were performed in COVID-19 patients at their center. However, 61 centers inserted chest tubes for pneumothorax and 56 for pleural effusion. Twenty-four centers performed tracheostomies. In ten centers, chest tubes were inserted to treat empyema (*n* = 5) or hemothorax (*n* = 5). Three centers performed surgical interventions for subcutaneous emphysema. Three centers performed bronchoscopies. Three centers reported performing surgery for tracheostomy-related complications and five for other causes.

**Question** **31.**
*Has the pandemic affected the management of pleural fluid drainage? (RR: 94/94; 100%)*


In 69 centers (73.4%), the management of pleural fluids was unchanged. However, 20 respondents (21.3%) reported that a higher proportion of patients were discharged to home with the drainage catheter still in place. Five centers (5.3%) did not discharge patients to home with the catheter in place.

**Question** **32.**
*Have the criteria for the drainage tube removal at your department been modified? (RR: 94/94; 100%)*


Most respondents (89.4%) maintain the same criteria for chest tube removal. However, 6.4% and 4.3%, respectively, reported removing the chest tubes either earlier or later than usual.

**Question** **33.**
*If the patient is discharged to home with a chest tube, what type of system do you use? (RR: 86/94 91.5%)*


Most centers (*n* = 25, 29.1%) used a digital system, followed by dry drains (*n* = 34, 35.4%), Heimlich valves (*n* = 14, 16.3%), and collection bags (*n* = 13, 15.1%).

### 3.6. Personal Protection

Table 4 summarises the findings regarding personal protection.

### 3.7. Teaching and Research

The survey results for teaching and research are shown in Table 5.

## 4. Discussion

The findings of this survey of thoracic surgeons reveals the impact of the COVID-19 pandemic on the surgical treatment of patients with lung cancer in Spain. Our data show that the pandemic had a major impact on clinical activity in thoracic surgery departments in Spain, one of the countries most affected by the first wave of COVID-19. The high incidence of COVID-19 in our country substantially increased demand for health care services, overwhelming the capacity of many hospitals within the Spanish National Health System, with a direct negative impact on mortality rates. Nearly half (47.3%) of the survey respondents reported that their hospital had to add ICU and hospital beds, and many were also obliged to refer patients to external facilities (field hospitals, hotels, fairgrounds), which were made available to meet the health care demand. This near collapse of the health care system was confirmed by the survey respondents, three-quarters of whom considered the situation in their hospital as either “very serious” (50% of respondents) or “critical” (25.5%). In this context, the Spanish Association of Surgery (AEC) developed a classification system with five alert levels to adjust surgical recommendations according to variations in the epidemiological status of the country over time [20].

The overwhelming demand for hospital care during the pandemic had a direct negative impact on the management of patients with a confirmed or suspected diagnosis of lung cancer. The results of our survey clearly reflect the impact of the pandemic on the diagnostic/therapeutic process in these patients. For example, most respondents (60%) reported delaying (or omitting) many preoperative tests due to the pandemic. In addition, some preoperative studies—such as pulmonary function testing or diagnostic procedures such as bronchoscopy or CT-guided transthoracic biopsy—were deferred or directly obviated in certain cases, a strategy that was considered unavoidable due to the epidemiological situation at that time.

Several proposals were made to mitigate the effects of the pandemic. In May 2020, the Society for Advanced Bronchoscopy published recommendations on performing bronchoscopies and airway management in patients during the epidemic, with an emphasis on stratifying outpatient bronchoscopies according to the patient’s clinical diagnosis and the urgency of the procedure. In patients with early-stage, resectable lung nodules or masses, those recommendations called for performing outpatient bronchoscopy within two weeks [21]. By contrast, in a survey conducted by the Asian Society for Cardiovascular and Thoracic Surgery, nearly three-fourths (73%) of respondents recommended omitting perioperative bronchoscopy, endobronchial ultrasound (EBUS), and electromagnetic navigation-guided bronchoscopy due to the elevated risk of infection during the procedure [22]. Some international clinical guidelines recommended performing only critical respiratory function tests (e.g., pulmonary function tests prior to lung resection surgery) while avoiding all non-essential tests due to the risk of exposure to the virus [23].

The pandemic also had a direct effect on meetings of the MTB, which in turn impacted decision-making in patients with lung cancer. Interestingly, despite the high hospital demand and case overload, only 3% of participating centers in this survey reduced the frequency of MTB meetings. This finding is worth highlighting given that the outsized impact of the virus in Spain compared to many other European countries. In fact, a survey carried out by the European Society of Thoracic Surgery (ESTS) found that 20% of MTB meetings were cancelled in member countries versus only 3% in Spain [24]. One of the reasons for the low cancelation rate in Spain could be the rapid uptake of online meetings (one-third of which were held online), indicating a rapid capacity for adaptation or perhaps the existence of previously-established digital logistics plan. In this regard, it would be interesting to determine if the proportion of virtual meetings has increased significantly over time. Several international recommendations emphasise the key role that the MTB plays in ensuring the optimal treatment of patients with thoracic malignancies during the pandemic [25].

Approximately two-thirds of the survey respondents reported that the mean waiting time for surgery in patients with a diagnosis or high suspicion of lung cancer was longer than one month. Some respondents suggested that implementation of special “COVID-free” hospitals could have decreased wait times. These long waiting times, attributable to the exceptional circumstances of the pandemic, led some institutions to consider alternatives to surgery. For example, 10% of respondents offered stereotactic body radiotherapy (SBRT) to patients with early-stage disease and nearly 30% of respondents reported a decrease in surgical resections for locally-advanced lung cancer. These changes in standard clinical practice may be at least partially attributable to preliminary data published around the time that this survey was performed suggesting a high rate of postoperative morbidity and mortality in complex COVID-19 patients. In fact, this would explain why surgeons initially avoided or delayed performing high-risk procedures, including neoadjuvant treatments. This reduction in surgical resection rates is slightly higher than the figures reported in the ESTS survey [24]. The absence of clear recommendations on elective and urgent thoracic surgery underscores the crucial role of MTBs in decision-making [17].

Given the increased risk of postoperative complications in COVID-19-positive patients, preoperative screening programs for SARS-Cov-2 were implemented for all patients scheduled for surgery. In fact, nearly all of the respondents indicated that preoperative PCR testing was considered essential in surgical patients. The relative importance of the various clinical and radiological screening tests has been studied [20] and it is clear that preoperative molecular testing should be performed as close as possible to the surgical intervention. However, the criteria used to determine patient eligibility for surgery after a positive test for COVID-19 (symptomatic or not) has not yet been established. Nonetheless, the available data suggests that delaying surgery in asymptomatic patients with a positive PCR test does not appear to be associated with higher postoperative morbidity and mortality rates [26].

The shortage of hospital resources (both material and human) due to the combination of high demand and reduced health care services during the first months of the pandemic made it necessary to prioritise patients on thoracic surgery waiting lists, with cancer patients (particularly those with advanced disease) given first priority. On 4 April 2020, the Thoracic Surgery Outcomes Research Network published consensus recommendations for triaging surgery in patients with thoracic malignancies [16], representing an important effort by the international thoracic surgery community to improve decision-making during the pandemic. This consensus statement classified thoracic malignancies and specific surgical procedures into three independent categories: priority, potentially-deferrable, and deferrable.

In our survey, 41.5% of the respondents indicated that, during the pandemic, surgeries were restricted to patients with cancer. More than one-third of these professionals (35.5%) followed the recommendations of the aforementioned consensus statement [16], with most of the others taking a common-sense approach to decision making. In July 2020, the International Association for the Study of Lung Cancer (IASLC) published detailed guidelines for the management of patients with lung cancer [27]. The Spanish Society of Thoracic Surgery, based on data from various international consensus statements and on the preliminary results of the present survey, published its own recommendations for triaging patients with thoracic neoplasms in December 2020 [17]. Our survey showed that confidence in the published recommendations was high, with close to 80% considering these recommendations to be of moderate to high scientific quality, despite being based on very early data.

Since a high percentage of thoracic surgeries are performed in cancer patients, most of whom are high-risk, it can be difficult to strike a proper balance between the indication for surgery versus the risk of exposure to SARS-CoV-2, especially given that some published reports suggested that mortality rates in COVID-19-positive patients may be up to 30 times greater than patients without COVID-19, with a similarly elevated risk of pulmonary complications [28]. Nevertheless, in our survey, most respondents (71%) did not report high morbidity and mortality rates, which is especially remarkable considering that this patient cohort was largely comprised of non-deferrable cancer patients. Furthermore, only 11% attributed worse outcomes to COVID-19.

The negative impact of deferring surgical resection in patients with lung cancer is well known. Therapeutic delay has been shown to reduce survival rates, even in patients with early-stage disease (starting with stage IA2 adenocarcinoma and stage IB squamous carcinoma) [29]. As the survey was performed during the first phase of the pandemic, we do not have information about how any delays in diagnosis or detection of lung cancer might have affected the disease in terms of size, stage, or eligibility for curative surgery. However, previously published studies have demonstrated that cancer screenings, visits, treatment, and surgery have decreased by up to 70% [30]. As a result, cancer morbidity and mortality is expected to increase for years to come. Given the need to postpone surgery in these patients during the pandemic, some authors proposed SBRT as an alternative treatment in patients with early-stage lung cancer as a “bridge” until surgery could be performed [31]. However, to date, no clear guidelines have been proposed regarding the indications for these alternative therapeutic strategies—which are associated with worse outcomes—in part because the use of these alternative approaches require an individualised assessment by the MTB. In the context of the current pandemic, some reviews have proposed omitting surgery in resectable, locally-advanced lung cancer; however, in our view as surgeons, this would deprive the patient of the most effective therapeutic strategy [32].

This study has several limitations, including possible biases related to the questionnaire design and wording. In addition, our sample was mainly comprised of thoracic surgeons at Spanish hospitals, which limits the sample size and does not provide a multidisciplinary perspective. Only limited data are available to assess the true impact of the pandemic on surgical activity and outcomes for NSCLC in Spain. The GRAVID study carried out by the Spanish Lung Cancer Group offers a general perspective about patients with lung cancer and COVID-19, but it did not evaluate surgical issues [33]. Nevertheless, the survey provides a global perspective of the situation in thoracic surgery departments during the COVID-19 pandemic, and therefore of the surgical treatment of lung cancer during the initial and most critical time points in one of the countries most affected by the virus. Moreover, it allows us to compare these results with those obtained by other countries to identify similarities and differences. Anyway, to ascertain the true impact of the pandemic on lung cancer mortality rates, multicenter registries (ideally national registries) are needed to accurately determine the impact of the reduction in surgical procedures on the therapeutic options offered to patients.

## 5. Conclusions

The present survey provides key data on the response of thoracic surgery departments in Spain to the severe—albeit heterogeneous—impact of COVID-19 on the management of lung cancer patients. The findings of this survey underscore the resilience of these professionals during the pandemic, who made every effort to provide these cancer patients with the best treatment possible while minimising the risk of exposure to SARS-CoV-2. The initial lack of common protocols at the onset of the pandemic led to a wide range of strategic responses, with a decision-making supported by the experience of multidisciplinary teams.

## Figures and Tables

**Figure 1 cancers-13-02897-f001:**
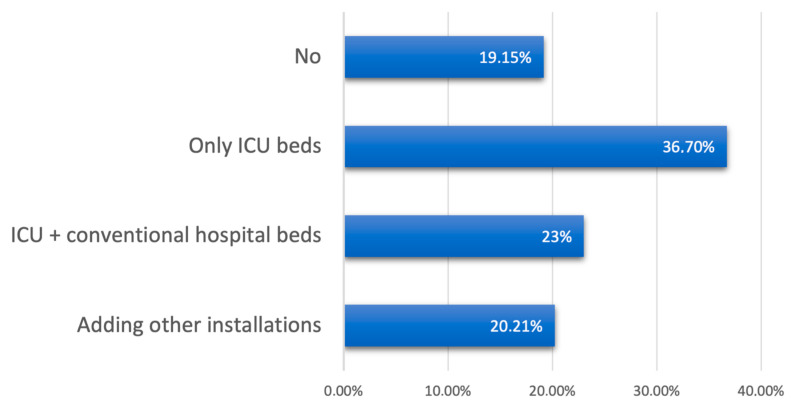
Did your hospital have to increase the number of beds? How?

**Figure 2 cancers-13-02897-f002:**
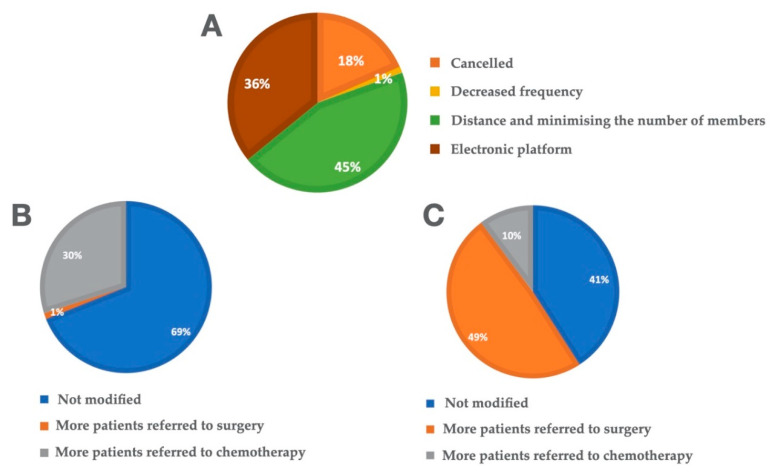
Multidisciplinary tumour boards. (**A**) Was it possible to maintain the routine work of the multidisciplinary tumour board at your hospital? (**B**) In patients with early-stage lung cancer, did the pandemic alter the treatment decisions made by the MTB. (**C**) In patients with locally-advanced lung cancer, did the pandemic alter the treatment decisions made by the MTB?

**Table 1 cancers-13-02897-t001:** Survey: questions and options.

Sociodemographics and Hospital-Related Data
1. What country do you work in?
Spain Other
2. What type of institution do you work at?
University Public hospital Non-University Public hospital Private hospital University Private hospital
3. How many beds does your hospital have?
100 beds or less Between 100 and 400 beds Between 400 and 800 beds >800 beds
4. Did your hospital have to increase the number of beds?
No Yes, only for ICU beds Yes, it was necessary to increase hospital beds Yes, it was necessary to use other installations (gym, other centers, …)
5. Was it necessary to adapt other spaces in the hospital to treat COVID-19 patients (e.g., gymnasium, library, cafeteria, field hospital, etc.)?
Yes No
6. Did you refer patients from your center to hotels or other facilities?
Yes No
7. How affected was the normal functioning of your hospital?
Not at all Slightly (some minor elective surgeries has been posponed, not other changes) Moderately (Significant emergency department activity, more surgeries postponed, less ICU beds available for non-COVID-19 patients) Severe (significant emergency department activity, only medically or oncologically urgent surgeries are executed, minority of ICU beds available for non COVID-19 patients) Extreme (hospital care is insufficient for this pandemic: shortness of beds, staff, supplies and ICU resources)
8. Have healthcare staff been tested for SARS-CoV-2?
No Yes, everyone Yes, but only after having symptoms Yes, only after exposure to COVID-19 patient Yes, after having symptoms or exposure to COVID-19 patient The criteria is not well defined
9. Has any member of the department been tasked with treating patients admitted to the inpatient ward for COVID-19?
No Yes, but only a few Yes, all members
**Multidisciplinary Teams and Lung Cancer**
10. Was it possible to maintain the routine work of the multidisciplinary tumour board at your hospital?
No, they’ve been suspended Yes, but with social distancing and minimizing the number of assistants Yes, using e-platform Yes, but less frequently
11. In patients with early-stage lung cancer, did the pandemic alter the treatment decisions made by the MTB?
Not at all Scheduled for surgery only A bigger number are scheduled for radiotherapy (SBRT) Patients transferred to other centers for surgery
12. In patients with locally-advanced lung cancer, did the pandemic alter the treatment decisions made by the MTB?
Not at all More patients have been scheduled to surgery More patients have been scheduled for systemic therapy Patients transferred to other centers for surgery
13. What is your opinion regarding the changes in the management of patients with lung cancer?
I agree on how it is being acted I think other centers should be designated as non-COVID-19 in order to maintain surgical activity I think other areas in the same hospital should be designated as non-COVID-19 in order to maintain surgical activity
14. Since the start of the pandemic, what is the average waiting time for surgery in your cancer patients?
<1 month 1–2 months 2–3 months >3 months
**Screening**
15. Did you preoperatively test for SARS-CoV-2 in patients scheduled to undergo thoracic surgery?
Every patient is screened. This test is necessary before going into the OR None Only symptomatic patients
16. In patients admitted for thoracic surgery, what was the indication to perform a preoperative diagnostic test for SARS-CoV-2?
None Every patient Only symptomatic patients
17. What type of diagnostic tests for SARS-CoV-2 are routinely performed at your center?
Nasopharyngeal swab Sputum Blood test for antibodies Bronchoalveolar lavage Chest X-Ray Chest CT
18. In your opinion, when should screening for SARS-CoV-2 infection be performed?
Every patient before surgery Every patient after surgery Only when symptomatic
19. In patients with a positive preoperative SARS-CoV-2 test result, does this influence surgical planning in any way?
The patient goes through surgery. Health worker wear necessary protection. Surgery will be postponed at least 14 days. Surgery will be postponed only if patient is symptomatic. Yes, surgery is suspended and referred for alternative treatment.
**Preoperative Workup**
20. How has the COVID-19 pandemic influenced the preoperative workup?
Not at all, all investigations are available in a normal time frame Only pet-CT investigations are delayed, or unavailable Only endobronchial investigations (bronchoscopy, EBUS) are delayed or unavailable CT guided biopsy is not routinely available Pneumology consultation (lung nodule study) is delayed or unavailable
21. Which of the following preoperative lung function tests (pulmonary function testing, pulmonary diffusion test, …) are not available due to the COVID-19 pandemic?
All investigations are available as normal Spirometry with or without arterial blood analyses and DLCO Cardiopulmonary exercise test (CPET) V/Q scan
22. What is your opinion with regard to changes in the preoperative workup?
I agree according to the situation We should minimize the number of tests We should do more tests
23. How has the pandemic affected consultations in thoracic surgery?
Nothing, everything continues with the same operation Everything has been suspended The face-to-face visits of new patients and the first post-operative visits are maintained. The rest is done electronically. All consultations are made electronically
**Surgery**
24. How has the pandemic affected surgical activity in your department?
Everything remains the same Everything has been suspended Benign pathologies have been discontinued. The rest have not been altered Exclusively operated on tumors with priority class I (American College of Surgeons classification) https://www.facs.org/covid-19/clinical-guidance/elective-case/thoracic-cancer (accessed on 20 February 2021) Only emergencies are operated
25. What is your opinion with regard to these changes in surgical interventions?
I agree I agree but I think we should further reduce surgical activity I do not agree. Under these conditions, the care of my patients is suboptimal I do not agree. An effort should be made to operate on more patients Other
26. Has the postoperative length of stay in the ICU/recovery unit been affected by the pandemic?
No Yes. The stay in reanimation unit has been reduced Yes. Patients do not stay in reanimation unit and go directly to the ward
27. Were any of the patients admitted to your department (regardless of surgical status) diagnosed with SARS-CoV-2?
None One case <5 cases >5 cases
28. What recommendations did you use for surgical planning in your department?
The classification proposed by the American College of Surgeons None in particular. We act according to availability and common sense There is nothing to prioritize since everything has been suspended The links on the SECT website have been very useful for decision-making Other (specify)
29. If surgical activity has continued at your hospital, have you observed any increase in morbidity and/or mortality?
No Yes, possibly attributed to the shortage of material and human resources Yes, but I don’t think it is related to the COVID-19 epidemic Yes, the morbidity and mortality has increased because of COVID-19 Other (specify)
30. What types of surgical interventions (if any) have been performed in your department on COVID-19 patients?
Thoracic drains due to pneumothorax Thoracic drains due to pleural effusion Tracheostomies Other (specify)
31. Has the pandemic affected the management of pleural fluid drainage?
No, is still the same Yes, we have increased home discharges with drainage Yes, we do not discharge anyone with drainage
32. Have the criteria for the drainage tube removal at your department been modified?
No Yes, we remove drains sooner Yes, we remove drains later
33. If the patient is discharged to home with a chest tube, what type of system do you use?
Digital device Collection bag Dry drain Other (specify)
**Personal Protection**
34. What type of protections are used in surgical procedures for patients who have not been tested for SARS-CoV-2 or whose status is unknown?
Standard measures FFP2/FFP3 and face shields for all attendees FFP2/FFP3 and face shields for surgeons. Complete PPE for the anesthesiologist Only when it comes to an airway opening procedure, complete PPE for everyone Everyone with full PPE
35. What types of protections are used in surgical procedures for patients who test negative for SARS-CoV-2?
Standard measures FFP2/FFP3 and face shields for all attendees FFP2/FFP3 and glasses for surgeons. Complete PPE for the anesthetist Only when it comes to an airway opening procedure, complete PPE for everyone Everyone with full PPE
36. What types of protections are used in surgical procedures performed in patients who test positive for SARS-CoV-2?
Standard measures FFP2/FFP3 and glasses for all attendees FFP2/FFP3 and glasses for surgeons. Complete PPE for the anesthetist Only when it comes to an airway opening procedure, complete PPE for everyone Everyone with full PPE
37. Among the department staff, what percentage of members have tested positive for SARS-CoV-2?
None <25% 25–75% >75%
**Teaching and Research**
38. What is your opinion with regard to the quality of the studies published to date?
Low quality, written too fast Moderate scientific quality High scientific quality, taking into account the situation
39. How many articles about SARS-CoV-2 and/or COVID-19 infection have you read?
None <5 5–10 >10
40. If you are a resident, how has the pandemic has affected your training?
It has affected me positively It has not affected me It has affected me negatively

**Table 2 cancers-13-02897-t002:** Survey results: Sociodemographic and hospital-related data.

Question	N	%
What country do you work in?		
Spain	90	96.8
Other	3	3.2
	Total 93	
What type of institution do you work at?		
University Public Hospital	88	89.1
Non-University Public Hospital	1	2.1
Private Hospital	7	8.1
University Private Hospital	3	4.1
	Total 94	
How many beds does your hospital have?		
≤100	3	3.2
100 to 400	12	12.8
400 to 800	36	38.3
>800	43	45.7
	Total 94	
Did your hospital have to increase the number of beds?		
No	18	19.1
Yes, but only for ICU beds	34	36.2
Yes	23	24.5
Yes, by adding beds at other facilities (gym, other centers, …)	19	20.2
	Total 94	
Was it necessary to adapt other spaces in the hospital to treat COVID-19 patients (e.g., gymnasium, library, cafeteria, field hospital, etc.)?		
Yes	56	59.6
No	38	40.4
	Total 94	
Did you refer patients from your center to hotels or other facilities?		
Yes	44	46.8
No	49	52.1
	Total 94	
To what extent was the normal functioning of your hospital affected?		
Not at all	1	1.1
Slightly	6	6.4
Moderately	16	17.0
Severely	47	50.0
Extremely	24	25.5
	Total 94	
Have healthcare staff been tested for SARS-CoV-2?		
No	4	4.3
Yes, everyone	19	20.2
Yes, but only after developing symptoms	18	19.1
Yes, only after exposure to a COVID-19 patient	3	3.2
Yes, after presenting symptoms or exposure to COVID-19 patient	22	23.4
The criteria are not well defined	31	33.0
	Total 94	
Has any member of the department been tasked with treating patients admitted to the inpatient ward for COVID-19?		
None	56	59.6
Only a few members of the department	23	24.4
All members	15	16.0
	Total 94	

**Table 3 cancers-13-02897-t003:** Survey results on “COVID-19 patient screening” section.

Question	N	%
Did you preoperatively test for SARS-CoV-2 in patients scheduled to undergo thoracic surgery?		
Every patient is screened.	91	96.8
None	2	2.1
Only if symptomatic	1	1.1
	Total 94	
In patients admitted for thoracic surgery, what was the indication to perform a preoperative diagnostic test for SARS-CoV-2?		
All patients	39	41.9
None	3	3.2
Only if symptomatic	51	54.8
	Total 93	
What type of diagnostic tests for SARS-CoV-2 are routinely performed at your centre?		
Nasopharyngeal swab	92	98.9
Sputum	0	0.0
Serology	17	18.3
Bronchoalveolar lavage	0	0.0
Chest X-ray	23	24.7
Thorax CT	14	15.1
	Toal 93	
In your opinion, when should screening for SARS-CoV-2 infection be performed?		
In all patients before surgery	92	98.9
In all patients after surgery	1	1.1
Only in symptomatic cases	0	0.0
	Total 93	
In patients with a positive preoperative SARS-CoV-2 test result, does this influence surgical planning in any way?		
No modification of surgical plan	4	4.0
Postpone surgery >14 days	82	88.0
Postpone only if symptoms	5	5.0
Surgery is suspended and patient is referred for alternative treatment	2	2.0
	Total 93	

**Table 4 cancers-13-02897-t004:** Survey results: Personal protection.

Question	N	%
What type of protections are used in surgical procedures for patients who have not been tested for SARS-CoV-2 or whose status is unknown?		
Standard measures	15	16.3
FFP2/FFP3 and face shields for the entire surgical team	43	46.7
FFP2/FFP3 and face shields for surgeons. Complete PPE for the anesthesiologist	16	17.4
Only for airway opening procedures. Complete PPE for everyone.	10	10.9
Everyone with full PPE	22	23.9
	Total 92	
What types of protections are used in surgical procedures for patients who test negative for SARS-CoV-2?		
Standard measures	44	46.8
FFP2/FFP3 and face shields for all attendees	42	44.7
FFP2/FFP3 and face shields for surgeons. Complete PPE for the anesthesiologist	9	9.6
Only for airway opening procedures. Complete PPE for everyone.	5	5.3
Everyone with full PPE	2	2.1
	Total 94	
What types of protections are used in surgical procedures performed in patients who test positive for SARS-CoV-2?		
Standard measures	0	0.0
FFP2/FFP3 and face shields for the entire surgical team	6	6.5
FFP2/FFP3 and face shields for surgeons. Complete PPE for the anesthesiologist	12	13.0
Only for airway opening procedures. Complete PPE for everyone.	3	3.3
Everyone with full PPE	73	79.3
	Total 92	
Among the department staff, what percentage of members have tested positive for SARS-CoV-2?		
None	45	48
<25%	17	18
25–50%	21	22
50–75%	9	10
>75%	2	2
	Total 94	

PPE: Personal Protective Equipment.

**Table 5 cancers-13-02897-t005:** Survey results: Teaching and research.

Question	N	%
What is your opinion with regard to the quality of the studies published to date?		
Low quality, written too fast	20	21.5
Moderate scientific quality	51	54.8
High scientific quality given the situation	22	23.7
	Total 93	
How many articles about SARS-CoV-2 and/or COVID-19 infection have you read?		
None	1	1.1
<5	13	13.8
5–10	30	31.9
>10	50	53.2
	Total 94	
If you are a resident, how has the pandemic has affected your training?		
It has affected me positively	3	12.5
It has not affected me	9	37.5
It has affected me negatively	12	50.0
	Total 24	

## Data Availability

The data presented in this study are available in this article.

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
