# Peer review of "Effect of COVID-19 on Thoracic Oncology Surgery in Spain: A Spanish Thoracic Surgery Society (SECT) Survey"

_cancers, 2021, doi:10.3390/cancers13122897_

Round 1
Reviewer 1 Report
The authors show the impact of the COVID-19 pandemic on thoracic oncology surgery in Spain. This manuscript is well written and should be published for the world to see now. There is an urgent need for thoracic surgeons to report on the perioperative management of lung cancer patients under these current challenging conditions. A few comments by me on this manuscript would be of interest to the readers, mainly thoracic surgeons.
Regarding question 12, it is probably oncologists or respiratory physicians who give chemotherapy to patients with advanced lung cancer. Are there more cases where surgeons have to give chemotherapy instead? Respiratory physicians are probably too busy with the COVID-19 pandemic to be involved in chemotherapy.
In the discussion, will there be more cases where surgeons decide to operate without tissue diagnosis of lung nodules?
The author mentions in great depth the effect of COVID-19 on lung cancer. What effect does it have on mediastinal tumors such as thymoma?
Author Response
RESPONSE TO REVIEWER 1
The authors show the impact of the COVID-19 pandemic on thoracic oncology surgery in Spain. This manuscript is well written and should be published for the world to see now. There is an urgent need for thoracic surgeons to report on the perioperative management of lung cancer patients under these current challenging conditions. A few comments by me on this manuscript would be of interest to the readers, mainly thoracic surgeons.
Regarding question 12, it is probably oncologists or respiratory physicians who give chemotherapy to patients with advanced lung cancer. Are there more cases where surgeons have to give chemotherapy instead? Respiratory physicians are probably too busy with the COVID-19 pandemic to be involved in chemotherapy.
Thank you very much for your suggestions and contributions. We agree with your opinion regarding the need for a greater understanding of the effects of the COVID-19 pandemic on lung cancer treatment. As thoracic surgeons, we were especially interested in the impact on surgical treatment. In fact, our aim was to perform this survey just as the first (and worst) wave was still underway.
Regarding your question, our survey did not consider the possibility that physicians other than medical oncologists administered chemotherapy since, in our country, medical oncologists are the only physicians accredited to prescribe and administer chemotherapy. Neither respiratory physicians nor surgeons can prescribe these agents.
In the discussion, will there be more cases where surgeons decide to operate without tissue diagnosis of lung nodules?
We did not directly assess this question. However, based on responses to the other questions, we believe that this was, in fact, the case. There has been a 59.8% of preoperative workup affection, being bronchoscopy in 56.5% of the cases and CT-guided biopsy (42.4%) the most affected ones, while only 39.4% of the centers completely suspended their surgical activity. Seeing such different figures, we can then deduce that surely, there were more surgical patients without presurgical tissue diagnosis.
The author mentions in great depth the effect of COVID-19 on lung cancer. What effect does it have on mediastinal tumors such as thymoma?
Thank you for pointing out this oversight. Regrettably, other malignancies such as thymoma are beyond the scope of our survey. This was done intentionally, as our focus was on lung cancer. That , but it has surely been an important effect, at least, on the time to surgery, taking into account that the American College of Surgeons, which guidelines has been the gold-standard in decision-taking in our country, only recommended surgery in bulky and symptomatic mediastinal tumors.

Reviewer 2 Report
I compliment you on your interesting paper.
If possibile I would rather suggest to explore if Covid 19 caused a delayed access to surgery and - as a consequence - you operated on more advanced stages in that period (this is what happend in my personal experience).
You may assess the absolute numbers of early stage and advanced stage operated lung cancers during the pandemics and - moreover - you may compare these numbers with the 2019 data.
Author Response
RESPONSE TO REVIEWER 2
I compliment you on your interesting paper.
If possible I would rather suggest to explore if Covid 19 caused a delayed access to surgery and - as a consequence - you operated on more advanced stages in that period (this is what happened in my personal experience).
We thank the reviewer for the interesting comments and suggestions. With regard to your comment about delayed access to surgery, we address this issue in the revised discussion section. Please, see the last two paragraphs in the revised discussion section.
You may assess the absolute numbers of early stage and advanced stage operated lung cancers during the pandemics and - moreover - you may compare these numbers with the 2019 data.
To the best of our knowledge, only two centers in Spain have reported the specific numbers of oncologic surgical resections (references 18 and 19 in the manuscript). One of these centers found that, at the peak of the first wave, surgical activity had declined by 44% compared to the same period in 2019 (19). However, those authors did not report how many of these cases were early or advanced lung cancers, although they did specify that the cancer patients who were selected for surgery were those considered non-delayable.
In short, it is not possible to draw any definitive conclusions or specify a specific reduction in oncological surgical activity in the entire country.

Reviewer 3 Report
Attached

Author Response
RESPONSE TO REVIEWER 3
Comments for authors: Effect of COVID-19 on thoracic oncology surgery in one of its 2 most affected countries: a Spanish Thoracic Surgery Society 3 (SECT) survey.
Heading – Thoracic oncology surgery is far more comprehensive than the aspects touched in the paper. I suggest editing it. “The most affected country” can be commented in the paper, I think Spain is enough.
Thank you very much for this paramount contribution. We’ve done this change in the heading.
Introduction:
- A brief description of lung cancer situation in Spain before the pandemic
- Any available data on lung cancer in Spain at present or statement that information is limited.
Discussion:
- More national data about lung cancer management is required to illustrate great efforts of Spanish thoracic surgeons during the pandemic.
Thank you for these important comments. We acknowledge that there is limited data about lung cancer in Spain at present. The Spanish Society of Medical Oncology estimated that there were approximately 29,000 new lung cancer cases in Spain in 2020 (https://seom.org/dmcancer/cifras-del-cancer/). Another study calculated that cancer diagnoses (all organs) decreased by 20.8% from 2019 to 2020 (Amador M, Matias-Guiu X, Sancho-Pardo G, Contreras Martinez J, de la Torre-Montero JC, Peuelas Saiz A, Garrido P, García-Sanz R, Rodríguez-Lescure Á, Paz-Ares L. Impact of the COVID-19 pandemic on the care of cancer patients in Spain. ESMO Open. 2021 Apr 30;6(3):100157. doi: 10.1016/j.esmoop.2021.100157. Epub ahead of print. PMID: 34015642). Unfortunately, that study did not specify the numbers with regard to lung cancer cases. Therefore, it would be difficult to give detailed information about the lung cancer diagnosis and treatment situation.
- Establishing goals of the study in addition to the aim will be helpful. Eg. “How pandemic influenced the rate of curative surgery?”.
Thank you for this contribution. We have modified the study aims accordingly. Please see the last paragraph of the introduction.
Materials and Methods:
1. I think you sould present here the questionnaire itself, because it was the main tool of the study and you present it anyway in the Results.
Thank you very much for this suggestion. We initially considered sending it as supplementary material in order to avoid including too many details in the manuscript, but following your indications, we have included it. If you believe it would be better as supplementary material instead, please let us know.
- A short description how the questionnaire was designed will be appropriate. Eg the number of questions, expected time to fill it in etc.
Thank you for your comment. We have added this information to the second paragraph of ‘Material and Methods section’, where the number of questions are described by area of interest. We have also included the expected time needed to complete the survey.
Results:
- I recommend changing the “question-answer” style and to use plain text alone or in combination with tables. It will help to reduce repetitions.
- You may group the response rate eg 94/94 or 100% - 80 in a table or as a plain text.
Initially, the manuscript was written as you suggest. However, although this reduced repetitions, it also made the manuscript more difficult to understand clearly, which is why we opted for the question-answer style. Although that study was less “reader friendly”, it was easier to understand and to find the relevant information (due to the systematic structure). In any case, as you recommend, to limit the large amount of highly repetitive text, we have converted the less important questions into tables, as you will see in the revised version of the text.

Round 2
Reviewer 3 Report
No comments